# Working lives of GPs in Scotland and England: cross-sectional analysis of national surveys

Helen Hayes,[1] Jonathan Gibson,[1] Bridie Fitzpatrick,[2] Kath Checkland,[1] Bruce Guthrie,[3] Matt Sutton,[1] John Gillies,[3] Stewart W Mercer [3]

We presented the findings of the study at the Scottish School of Primary Care conference on 7 May 2019.

[1]Health Organisation, Policy and Economics Research Group, Centre for Primary Care & Health Services Research, School of Health Sciences, University of Manchester, Manchester, UK
[2]Institute of Health and Wellbeing, University of Glasgow, Glasgow, UK
[3]Usher Institute, College of Medicine and Veterinary Medicine, The University of Edinburgh, Edinburgh, UK

**Correspondence to**
Professor Stewart W Mercer;
stewart.mercer@ed.ac.uk

## ABSTRACT

**Objectives** The UK faces major problems in retaining general practitioners (GPs). Scotland introduced a new GP contract in April 2018, intended to better support GPs. This study compares the career intentions and working lives of GPs in Scotland with GPs in England, shortly after the new Scotland contract was introduced.

**Design and setting** Comparison of cross-sectional analysis of survey responses of GPs in England and Scotland in 2017 and 2018, respectively, using linear regression to adjust the differences for gender, age, ethnicity, urbanicity and deprivation.

**Participants** 2048 GPs in Scotland and 879 GPs in England.

**Main outcome measures** Four intentions to reduce work participation (5-point scales: 1='none', 5='high'): reducing working hours; leaving medical work entirely; leaving direct patient care; or continuing medical work but outside the UK. Four domains of working life: job satisfaction (7-point scale: 1='extremely dissatisfied', 7='extremely satisfied'); job stressors (5-point-scale: 1='no pressure', 5='high pressure); positive and negative job attributes (5-point scales: 1='strongly disagree', 5='strongly agree').

**Results** Compared with England, GPs in Scotland had lower intention to reduce work participation, including a lower likelihood of reducing work hours (2.78 vs 3.54; adjusted difference=−0.52; 95% CI −0.64 to −0.41), a lower likelihood of leaving medical work entirely (2.11 vs 2.76; adjusted difference=−0.32; 95% CI −0.42 to −0.22), a lower likelihood of leaving direct patient care (2.23 vs 2.93; adjusted difference=−0.37; 95% CI −0.47 to −0.27), and a lower likelihood of continuing medical work but outside of the UK (1.41 vs 1.61; adjusted difference=−0.2; 95% CI −0.28 to −0.12). GPs in Scotland reported higher job satisfaction, lower job stressors, similar positive job attributes and lower negative job attributes.

**Conclusion** Following the introduction of the new contract in Scotland, GPs in Scotland reported significantly better working lives and lower intention to reduce work participation than England.

## INTRODUCTION

Low physician job satisfaction, stress and burnout are associated with increased staff turnover and threats to patient safety,[1–4] and contribute to recruitment and retention problems in general practice in the UK. Job

## Strengths and limitations of this study

► This is the first study to compare the working lives of general practitioners (GPs) in Scotland and England using a set of common, validated questions on satisfaction, job characteristics and job pressures using a similar data collection method.

► In both countries, largely nationally-representative samples were collected.

► We have adjusted the differences for respondents' age, gender, ethnicity and partner status, as well as the deprivation and urbanicity of the populations registered with the practices at which these GPs work.

► The response rate was lower to the England survey. Evidence suggests that less satisfied GPs are more likely to respond to job satisfaction surveys. If also true in this context, then the lower response rate in the English survey may mean it represents a pool of less satisfied English GPs and the differences could overestimate the true population differences.

► However, our supplementary analyses restricting the Scottish sample to the first 25.2% of returned surveys to match the total response rate in the English survey, showed only trivial differences compared with the full sample.

satisfaction of general practitioners (GPs) fell markedly in the UK between 2012 and 2015,[5] and in a recent survey of 11 countries, the job satisfaction of UK GPs was second lowest.[6] The general practice 'crisis'[7] in the UK reflects a substantial increase in workload[8] and surveys of the working lives of English GPs have shown an increasing proportion intending to leave direct patient care within 5 years, up to a high of 39% in 2017.[9]

Despite policy ambitions to increase the number of GPs in the UK, there was a 1.4% decrease in full-time equivalent GPs in England between September 2017 and September 2018.[10] Failure to address GP recruitment and retention puts a number of health policy goals at risk, including shifting care out of hospital and into the community.

Whereas, England has emphasised market mechanisms and inspection regimes to assure quality, Scotland has placed the emphasis on collaboration and integration. England retains the Quality and Outcomes Framework (QOF) and has implemented Care Quality Commission inspections. As the first step in negotiating and implementing a radical new GP contract in April 2018, Scotland abolished the QOF in 2016 and established GP clusters as a clear signal of a shift to a more collaborative way of working.[11 12]

GP job satisfaction is an important indicator of the impact of GP contractual reform. We compare this between GPs in England in 2017 and Scotland in 2018 using cross-sectional survey data.

## METHODS

### Design and setting

Data for England was from the 2017 National GP Work-life Survey,[9] using validated measures of satisfaction,[13] job pressure and attributes.[14] Data for Scotland was from the Scottish School of Primary Care survey and included the same validated measures.[15]

### Participants

The English survey questionnaire was first posted to a random sample of 4000 GPs in October 2017 with an option to complete an online version. Two reminders letters were sent in November and December 2017. Responses were returned by 996 GPs, representing 25.2% of the 3953 eligible respondents after excluding GPs who had died, retired, left the practice or returned blank questionnaires.[9]

The Scottish survey questionnaire was first mailed to all 4371 GPs in July 2018 which included the option of completing an online version, followed by two reminder paper mailings with further copies of the questionnaire in August and September 2018. Additionally, two further email reminders (with the link to the online version) were sent in mid and late August. The Scotland sample includes responses from 2465 GPs (56.4% response rate). Due to the higher response rate, we ran supplementary analyses which restricted the sample in Scotland to the first 25.2% of returned surveys, to match the response rate in England.

The final sample consisted of 2927 GPs (2048 GPs in Scotland and 879 in England) after removal of individuals with missing information on the outcome variables and covariates.

### Outcome variables

We measured intention of GPs to reduce work participation and four related domains of working lives; job satisfaction, job stressors, positive job attributes and negative job attributes. For each domain, we created an average score for all of the questions for each respondent. Individual components are shown in the online supplemental tables S1–S4.

### Intentions to reduce work participation

We measured intention of GPs (in the next 5 years) to reduce work hours, leave medical work entirely, leave direct patient care or to continue medical work but outside the UK, answered on a 5-point scale with 1 indicating 'none', 2 'slight', 3 'moderate', 4 'considerable' and 5 'high'. We consider each of these components as separate outcomes.

### Domains of working lives

The four domains of working lives were defined as follows:

Job satisfaction: Satisfaction with 10 different aspects of their job.[13] Ratings on a 7-point scale from 1 'extremely dissatisfied' and to 7 'extremely satisfied'. The 10 answers were then averaged (minimum possible score of 1 and maximum possible score of 7).

Job stressors: Pressure experienced from 13 job factors, rated on a 5-point scale; 1 'no pressure', 2 'slight pressure', 3 'moderate pressure', 4 'considerable pressure' and 5 'high pressure'. The 13 answers were averaged (minimum possible score of 1 and maximum possible score of 5).

Positive job attributes: Nine statements relating to 'positive' or desirable job aspects were rated on a 5-point scale where 1 indicates 'strongly disagree', 2 'disagree', 3 'neutral', 4 'agree' and 5 'strongly agree'. The nine answers were averaged (minimum possible score of 1 and maximum possible score of 5).

Negative job attributes: Four statements which relate to 'negative' or undesirable job aspects were rated on a 5-point scale where 1 indicates 'strongly disagree', 2 'disagree', 3 'neutral', 4 'agree' and 5 'strongly agree'. The four answers were averaged (minimum possible score of 1 and maximum possible score of 5).

### Covariates

We adjusted for combinations of gender and age (five categories of 25–34, 35–44, 45–54, 55–64 and 65+ years); ethnicity (Black, Asian or minority ethnic group vs White ethnic group); practice urbanicity (proportion of patients living in an urban area),[16] and practice deprivation (proportion of patients living in an area in the most deprived quintile using an adjusted Index of Multiple Deprivation for England) to ensure comparability with Scotland.[17]

### Summary statistics and representativeness

The age, gender and partner status of respondents were compared with census data for GPs in England in December 2017[18] and in Scotland from the Information Services Division in September 2018.[19] We also compared, separately for each country, the average proportions of patients in the most deprived quintile and in an urban area, in the practices of the respondent GPs and all GPs.[16 19–22] The Scotland sample was more representative than the England sample, with women, younger doctors and non-partners being less under-represented in Scotland. GPs from more deprived and more urban practices

were under-represented in the England sample (online supplemental table S5). We adjusted for these factors in the country comparisons.

## Statistical analysis

Linear regression models with heteroscedasticity-robust standard errors to calculate point estimates and 95% CIs for the differences of Scotland from England. In the adjusted models, age, gender, age and gender interaction, ethnicity, practice urbanicity and practice deprivation were included as covariates. We used average scores for the domains: job satisfaction, job stressors, positive job attributes and negative job attributes. The analysis was repeated on each separate question within these domains (online supplemental tables S1–S4). The questions were measured on ordered categorical scales and so we also checked the robustness of the adjusted differences to the use of ordered probit regression models and results were qualitatively similar.

## RESULTS

### Intentions to reduce work participation

GPs in Scotland reported lower likelihood of reducing work hours (2.78 vs 3.54; adjusted difference −0.52; 95% CI −0.64 to −0.41), continuing medical work outside of the UK (1.41 vs 1.61; adjusted difference −0.2; 95% CI −0.28 to −0.12), leaving direct patient care (2.23 vs 2.93; adjusted difference −0.37; 95% CI −0.47 to −0.27) and leaving medical work entirely (2.11 vs 2.76; adjusted difference −0.32; 95% CI −0.42 to −0.22) compared with England (table 1, figure 1).

### Domain scores

GPs in Scotland reported statistically significantly higher levels of satisfaction (5.27 vs 4.71; adjusted difference 0.50; 95% CI 0.42 to 0.59), lower levels of pressure (3.53 vs 3.86; adjusted difference −0.29; 95% CI −0.35 to −0.23), similar positive job attributes (3.19 vs 3.15; adjusted difference 0.01; 95% CI −0.04 to 0.06) and less agreement in response to negative job attributes (4.03 vs 4.30; adjusted difference −0.27; 95% CI −0.31 to −0.22) (table 2).

Scottish GPs had higher job satisfaction across all 10 satisfaction domains, lower job stress for 12 of the 13

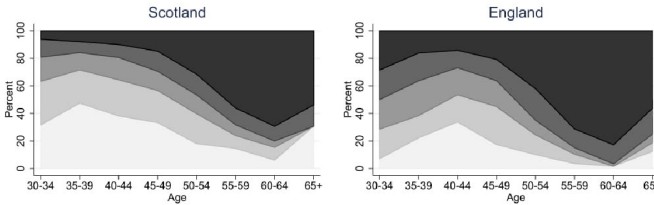

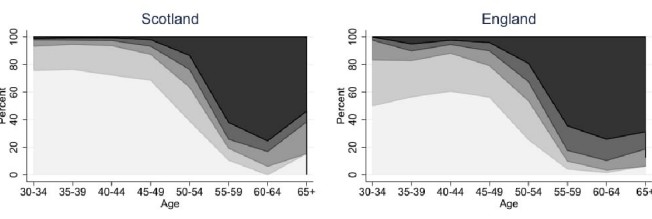

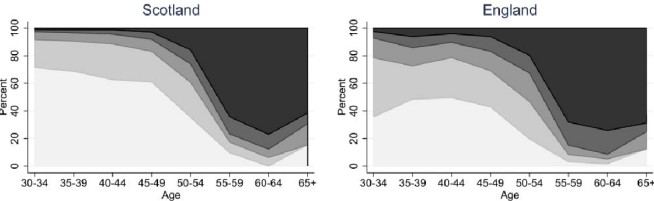

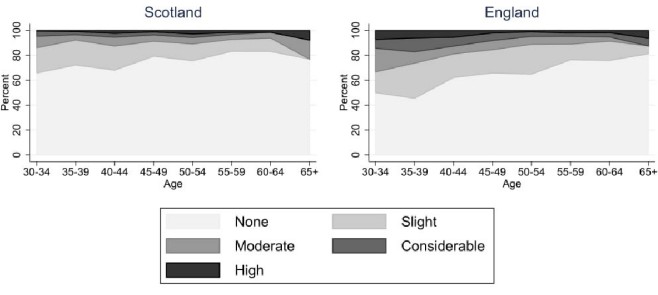

**Figure 1** General practitioners' intentions to reduce work hours in Scotland and England.

stressors (the exception being finding a locum), higher scores on six of the nine positive job attributes (the exceptions being involvement in changes affecting their work, flexibility in working time and clear feedback about performance) and lower scores on all four of the negative

| Table 1 | Differences between Scotland and England in intentions to reduce work participation | | | |
|---|---|---|---|---|
| | Scotland (mean, SD) | England (mean, SD) | Unadjusted difference (coef., 95% CI) | Adjusted difference (coef., 95% CI) |
| Reduce your work hours within 5 years? | 2.78 (1.56) | 3.54 (1.49) | −0.76 (−0.88 to −0.64) | −0.52 (−0.64 to −0.41) |
| Leave medical work entirely within 5 years? | 2.11 (1.49) | 2.76 (1.63) | −0.65 (−0.78 to −0.52) | −0.32 (−0.42 to −0.22) |
| Leave direct patient care within 5 years? | 2.23 (1.51) | 2.93 (1.59) | −0.70 (−0.82 to −0.58) | −0.37 (−0.47 to −0.27) |
| Continue with medical work but outside the UK within 5 years? | 1.41 (0.84) | 1.61 (1.03) | −0.21 (−0.28 to −0.13) | −0.20 (−0.28 to −0.12) |

N=2927. All four measures of intentions to reduce work participation are measured on a 5-point scale, 1='none' to 5='high'. Columns 4 and 5 are coefficients from linear regressions with heteroscedasticity-robust standard errors. Adjusted differences control for gender and age category interactions, ethnicity, partner status, and practice deprivation and urbanicity.

**Table 2** Differences between Scotland and England in satisfaction, job stressors and job attributes

| | Scotland (mean, SD) | England (mean, SD) | Unadjusted difference (coef., 95% CI) | Adjusted difference (coef., 95% CI) |
|---|---|---|---|---|
| Satisfaction | 5.27 (0.97) | 4.71 (1.07) | 0.55 (0.47 to 0.63) | 0.50 (0.42 to 0.59) |
| Job stressors | 3.53 (0.75) | 3.86 (0.66) | −0.34 (−0.39 to −0.28) | −0.29 (−0.35 to −0.23) |
| Positive job attributes | 3.19 (0.57) | 3.15 (0.62) | 0.037 (−0.011 to 0.085) | 0.013 (−0.038 to 0.063) |
| Negative job attributes | 4.03 (0.64) | 4.30 (0.57) | −0.27 (−0.32 to −0.23) | −0.27 (−0.31 to −0.22) |

N=2927. Satisfaction is measured from 1 (extremely dissatisfied) to 7 (extremely satisfied). Job stressors are measured from 1 (no pressure) to 5 (high pressure). Job attributes are measured from 1 (strongly disagree) to 5 (strongly agree) with 3 being neutral. Columns 4 and 5 are coefficients from linear regressions with robust standard errors. Adjusted differences control for age, gender, age×gender interaction, ethnicity, partner, and practice deprivation and urbanicity.

job attributes (figure 2). These differences remained in the adjusted regression analysis (online supplemental tables S1–S4)

In both countries, almost every job stressor was (on average) rated in the range moderate to considerable. GPs in both countries disagreed or were neutral (on average) in their rating of five positive job attribute statements, with only slight average agreement in relation to having an interesting variety to work, knowing their responsibilities, having a choice in how to do the job and being involved in changes introduced. On average, GPs in both countries agreed with all four negative job attribute statements (figure 2; online supplemental tables S1–S4).

All differences between England and Scotland remained robust after controlling for GP characteristics, and practice urbanicity and deprivation. In the supplementary analyses, which restricted the sample in Scotland to the first 25.2% of returned surveys to match the total response rate in the English survey, there were only trivial differences in the results compared with those using the full sample (online supplemental tables S6 and S7).

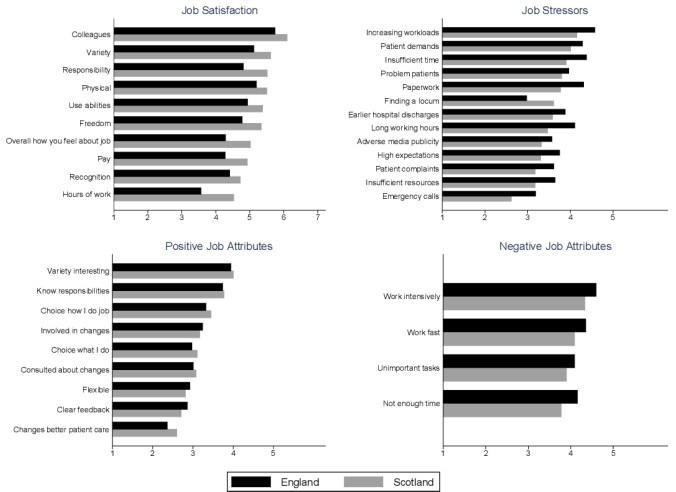

**Figure 2** General practitioners' mean responses to component questions of each domain in Scotland and England.

## DISCUSSION
### Summary
This study has demonstrated that compared with GPs in England, GPs in Scotland have lower intentions to reduce work participation, as well as higher levels of job satisfaction, lower job stressors and lower negative job attributes. These differences were of a reasonably large magnitude (one-third to one-half of SD), and thus likely to be meaningful in practice. It is possible that these differences relate, at least in part, to the recent changes in primary care in Scotland, including the new GP contract and the abolition (in April 2016) of the increasingly unpopular QOF.[23]

However, notwithstanding these differences, GPs in both countries, on average, rated their work as moderately to considerably stressful. In both countries, the average response to statements relating to positive job attributes clustered around 'neither agree nor disagree', and the average response to statements relating to negative job attributes clustered around 'agree'. Although Scottish GPs had less intention to reduce work participation than English, significant minorities in both countries intended to reduce their hours or other work participation in the next 5 years.

### Strengths and limitations
This is the first study to compare the working lives of GPs in Scotland and England using a set of common, validated questions on satisfaction, job characteristics and job pressures using a similar data collection method. In both countries, largely nationally-representative samples were collected. We have adjusted the differences for respondents' age, gender, ethnicity and partner status, as well as the deprivation and urbanicity of the populations registered with the practices at which these GPs work.

The response rate was lower to the England survey. Evidence suggests that less satisfied GPs are more likely to respond to job satisfaction surveys.[24] If also true in this context, then the lower response rate in the English survey may mean it represents a pool of less satisfied English GPs and the differences could overestimate the true population differences. However, our supplementary analyses restricting the Scottish sample to the first 25.2%

of returned surveys to match the total response rate in the English survey, showed only trivial differences compared with the full sample (online supplemental tables S6 and S7).

## Comparison with existing literature

The 2017 survey in England builds on previous English surveys on GP satisfaction using the same validated measures. This is the largest study ever on GPs' views on working lives in Scotland, and the first major study since the new Scottish GP contract was introduced in April 2018. Although Scotland generally had better scores than England, many of the absolute values in Scotland still give cause for concern. Almost one-quarter (23.8%) of GPs in Scotland rated their chance of leaving direct patient care as 'considerable' or 'high'. This compares with a small survey conducted by Royal College of General Practitioners, Scotland, in 2018 which found 26% of GPs felt they would be unlikely to be working in general practice in the next 5 years.[25] Thus, 'better' than England, does not necessarily mean 'good', and the findings give no room for complacency in either country.

A longitudinal study of GPs in the Wessex region of England conducted in 2014 and 2017 found worsening morale and higher intention to quit, with work intensity and amount being the most common reasons given.[26 27] Such longitudinal surveys are required in Scotland to understand the views of GPs as the new GP contract progresses, and to be able to compare with ongoing national surveys in England.

## Implications for research and policy

We feel it is plausible to assume that (at least some of) the differences between GPs in Scotland and England in the present surveys were due to the timing and contract differences. Responsiveness of job satisfaction to new contracts has been shown before in the UK,[28] although it is conceivable that differences between the two countries may have existed prior to the more recent policy changes; the new Scottish contract reflects longer-standing differences from England in relation to NHS organisation, performance management and professional-managerial relationships.[12]

In England, since the current survey was carried out, a new contract has been introduced in 2019 which focuses on the so-called 'primary care networks' (PCNs).[29] These are intended to cover between 30 000–50 000 people, although some are considerably larger.[30] The English contract also provides direct reimbursement for non-GP practitioners (eg, pharmacists, physiotherapists) working within the PCN; payment for participating in a PCN; and access to incentive funding. Practices are required to deliver structured medication reviews, more support for care homes, early diagnosis of cancer and anticipatory care for those most at risk of hospital admission. It is not yet clear what effect these changes will have on the workload and job satisfaction of GPs.[31] The initial draft of the service specifications[32] was rejected by GPs as a significant burden of extra work.[33 34]

Our study suggests that GPs in England feel additional pressures across all domains of practice compared with their Scottish colleagues. While working together collaboratively as a PCN may increase satisfaction by engendering a sense of local community and support, it seems likely that overall satisfaction will only increase if perceived negative job attributes and job stressors also reduce. It is therefore of vital importance to understand how the recruitment of non-GPs into general practice impacts on different types of work.

The COVID-19 crisis has clearly had a significant impact on general practice in both countries, with practices switching to a 'triage first' model with markedly reduced face-to-face contact with patients.[35] Some of these changes may endure, but, at present, it is not clear which ones. It will be some time before we can understand the impact of COVID-19 on the work, satisfaction and retention of GPs.

Further research is required to look at trends over time, as primary care reforms proceed across the UK in the coming years, and as the impact of COVID-19 evolves. In addition, research is required to understand how different elements of contract changes impact particular domains of satisfaction, and how workload changes in response to particular innovations.

**Acknowledgements** We would like to thank all the GPs in England and Scotland who took part in this study.

**Contributors** English survey: MS and KC devised the project. HH, MS and JG were involved in the planning and design of methods. JG, KC and MS were involved in the data collection for the English data. The data cleaning and analysis was carried out by HH. HH, MS, JG and KC were involved in the writing of the manuscript. Scottish survey: SWM, BG, JG and BF devised the project. BG, SWM, JG and BF were involved in the planning and design of the methods. BF led the data collection and data cleaning. HH carried out the data analysis. SWM led the revisions to the manuscript with input from BG. BF and JG read and revised the later versions. All authors read and agreed the final version prior to submission. The corresponding author attests that all listed authors meet authorship criteria and that no others meeting the criteria have been omitted.

**Funding** The GP Worklife Survey in England is commissioned by the Department of Health and Social Care and carried out by the Policy Research Unit in Health and Social Care Systems and Commissioning (PRUComm). PRUComm is funded by the National Institute for Health Research Policy Research Programme (Ref: PR-PRU-1217–20801). The GP Worklife Survey in Scotland was funded internally by the Scottish School of Primary Care.

**Disclaimer** The views expressed are those of the authors and not necessarily those of the National Institute for Health Research or the Department of Health.

**Competing interests** None declared.

**Patient consent for publication** Not required.

**Ethics approval** Ethical approval was granted for the English survey from the University of Manchester's Research and Ethics committee on 10 May 2017 (Ref: 2017-2638-3884).The Scottish survey was assessed as not requiring ethical approval by the National Health Service (NHS) or the University of Glasgow MVLS College Ethics Committee.

**Provenance and peer review** Not commissioned; externally peer-reviewed.

**Data availability statement** No additional data available.

**ORCID iD**
Stewart W Mercer http://orcid.org/0000-0002-1703-3664

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
