## [Reviewer comments · BMJ Open]

ARTICLE DETAILS

TITLE (PROVISIONAL)	Working lives of GPs in Scotland and England: cross-sectional analysis of national surveys
AUTHORS	Hayes, Helen; Gibson, Jonathan; Fitzpatrick, Bridie; Checkland, Kath; Guthrie, Bruce; Sutton, Matt; Gillies, John; Mercer, Stewart

VERSION 1 – REVIEW

REVIEWER	Katherine Owen Warwick Medical School, UK
REVIEW RETURNED	18-Aug-2020

GENERAL COMMENTS	The introduction of a Scotland version of the GP Worklife Survey is a welcome addition to a well validated longitudinal tool. However the inferences which the paper makes relating to the impact of the new Scottish contract are possibly premature (3 months in) and do not take into account other confounding factors relating to longstanding differences between General Practice in England & Scotland. To truly assess the impact of the contract would require a longitudinal approach in Scotland. There has been significant recent work in this area, particularly from the groups at Exeter & Warwick which is not referenced in the article but may support development of alternative hypotheses on the differences between English and Scottish GP satisfaction. A focus on understanding why the differences exist would be more pertinent. The National GP Worklife Survey in England is a useful and well validated tool for monitoring longitudinally the satisfaction of GPs & their career intentions, which can then be mapped against key changes in General Practice. The use of the same tool to survey GPs in Scotland is a welcome addition which may help further understanding of the reasons for GP dissatisfaction by exploring differences between the nations. This article clearly outlines the importance of GP satisfaction in terms of the impact on patient safety. The implied research question asks whether GP satisfaction and intention to remain working are more positive in Scotland, with the hypothesis that the model of “collaboration and integration” in the new Scottish contract as opposed to the model of “market forces and inspection” in England may be influencing this. There are two reasons why this hypothesis is problematic. Firstly, the direct comparison between Scottish & English General Practice has a number of confounding factors not considered in the paper. The two health services have been diverging since devolution of NHS services began in 1999, so the implication that it is the new contract that has made the difference is
---

	not demonstrated and would require a longitudinal approach within Scottish GPs. Even prior to devolution there have been key differences such as smaller average list size in Scotland which persist and may influence job satisfaction and intention to leave. Secondly, differences in timing and application of the questionnaire may have influenced results. The England survey was distributed in the autumn and winter, a time of year associated with increased workload, whereas the Scottish survey was in the summer months. The timing of the Scotland questionnaire only 3 months after the introduction of the new contract is perhaps too short a time period to be able to evaluate changes in job satisfaction associated with it- changes are often associated with an initial “bounce” which can then be lost as they bed in to practice. I would suggest the question of whether Scottish and English GPs show differences in job satisfaction is valid, but should be reframed as a baseline observation to be followed up longitudinally before clear inferences can be drawn. Inclusion of qualitative content would further the understanding of why these changes may exist. As the authors also point out, there is clearly a problem with job satisfaction in both countries- it is just less bad in Scotland. Methodologically the use of the validated tool and regressions to control for demographic factors is appropriate. The response rate is low in the England cohort, but the authors take steps to allow for this. There have been other large regional studies of GP satisfaction and intention to quit from Exeter and Warwick which I am surprised are not referenced as they have insights into what factors might be relevant from systematic review evidence and qualitative elements. These could support development of a hypothesis into why Scotland appears to be a more attractive place to work.
--	--

REVIEWER	Shamini Gnani Imperial College London, UK
REVIEW RETURNED	10-Sep-2020

GENERAL COMMENTS	Thank you for an interesting article, which I enjoyed reading. There are a few very minor edits, for the proof stage: page 6 line 52 - additional words used - " the response rate" page 7 line 47 - misspelling of relating Table 1 and 2 -would be good to use the same consistent wording wrt to adjusted differences for completeness in both tables
--

VERSION 1 – AUTHOR RESPONSE

Reviewer 1:

“The introduction of a Scotland version of the GP Worklife Survey is a welcome addition to a well validated longitudinal tool. However the inferences which the paper makes relating to the impact of the new Scottish contract are possibly premature (3 months in) and do not take into account other confounding factors relating to longstanding differences between General Practice in England & Scotland. To truly assess the impact of the contract would require a longitudinal approach in Scotland. There has been significant recent work in this area, particularly from the groups at Exeter & Warwick

which is not referenced in the article but may support development of alternative hypotheses on the differences between English and Scottish GP satisfaction. A focus on understanding why the differences exist would be more pertinent.”

Response: Thank you for raising these points, and the accompanying file in which you expand on these issues in more detail.

In terms of the impact of the new contract in Scotland being premature, the reviewer is correct in terms of formal contract implementation (the contract was published in early April 2018, and the survey began in July 2018, though wasn't completed until September 2018) but new contract evolution happens over several years. We have modified the introduction in paragraph 3 as follows;

“As the first step in negotiating and implementing a radical new GP contract in April 2018, Scotland abolished the QOF in 2016 and established GP Clusters as a clear signal of a shift to a more collaborative way of working.”

We have also made the interpretation more cautious in the first paragraph of the Discussion replacing 'it seems like' with 'it is possible' that observed differences are due to the new contract as shown below;

“It is possible that these differences relate, at least in part, to the recent changes in primary care in Scotland, including the new GP contract and the abolition (in April 2016) of the increasingly unpopular QOF.” (Page 11, line 5)

In terms of other possible factors that could explain the differences between the two surveys, although we alluded to other factors in the Discussion (page 12 para 3) we have now made the interpretation more cautious;

“We feel it is plausible assume (at least some of) the differences between GPs in Scotland and England in the present surveys were due to the timing and contract differences. Responsiveness of job satisfaction to new contracts has been shown before in the UK, although it is conceivable that differences between the two countries may have existed prior to the more recent policy changes; new Scottish contract reflects longer-standing differences from England in relation to NHS organisation, performance management and professional-managerial relationships.”

In terms of the papers published by the Exeter and Warwick group, we have now included two of these* in relation to the need for further longitudinal studies by adding a new paragraph on page 12;

“A longitudinal study of GPs in the Wessex region of England conducted in 2014 and 2017 found worsening morale and higher intention to quit, with work intensity and amount being the most common reasons given [new references here]. Such longitudinal surveys are required in Scotland, to understand GPs views as the new GP contract progresses, and to be able to compare with ongoing national surveys in England.” (Page 12, para 3).

*

Owen K, Hopkins T, Shortland T, and Dale J, (2019) GP retention in the UK : a worsening crisis. Findings from a cross-sectional survey. *BMJ Open*, 9 (2). e026048. doi:10.1136/bmjopen-2018-026048

Dale J, Potter R, Owen K, Parsons N, Realpe A, Leach J (2015) Retaining the general practitioner workforce in England : what matters to GPs? A cross-sectional study. *BMC Family Practice*, 16 (1). pp. 1-11. 140. doi:10.1186/s12875-015-0363-1

Reviewer: 2

“Thank you for an interesting article, which I enjoyed reading. There are a few very minor edits, for the proof stage:

page 6 line 52 - additional words used - " the response rate"

page 7 line 47 - misspelling of relating Table 1 and 2 -would be good to use the same consistent wording wrt to adjusted differences for completeness in both tables.”

Response: Thank you. We have corrected the additional words on page 6. Page 7 has no mention of Tables 1 and 2, so we are unsure what wording you would like us to correct. We are happy to do so if the editor can guide us as to what is required.

VERSION 2 – REVIEW

REVIEWER	Dr Kate Owen Warwick Medical School
REVIEW RETURNED	06-Oct-2020
GENERAL COMMENTS	Thank you for making the changes to the manuscript which now reads in a more balanced tone. I look forward to seeing how results from the surveys develop in the coming years.